# Strength Training in Swimming

**DOI:** 10.3390/ijerph19095369

**Published:** 2022-04-28

**Authors:** Klaus Wirth, Michael Keiner, Stefan Fuhrmann, Alfred Nimmerichter, G. Gregory Haff

**Affiliations:** 1Faculty of Training and Sports Sciences, University of Applied Sciences Wiener Neustadt, 2700 Wiener Neustadt, Austria; alfred.nimmerichter@fhwn.ac.at; 2Department of Sport Science, University of Health and Sports, 85737 Ismaning, Germany; michaelkeiner@gmx.de; 3Olympic Training and Testing Centre Hamburg/Schleswig-Holstein, 22049 Hamburg, Germany; s.fuhrmann@osphh-sh.de; 4School of Medical and Health Sciences, Edith Cowan University, Joondalup, WA 6027, Australia; g.haff@ecu.edu.au

**Keywords:** resistance training, start performance, turn performance, preventive training, elite swimmers

## Abstract

This narrative review deals with the topic of strength training in swimming, which has been a controversial issue for decades. It is not only about the importance for the performance at start, turn and swim speed, but also about the question of how to design a strength training program. Different approaches are discussed in the literature, with two aspects in the foreground. On the one hand is the discussion about the optimal intensity in strength training and, on the other hand, is the question of how specific strength training should be designed. In addition to a summary of the current state of research regarding the importance of strength training for swimming, the article shows which physiological adaptations should be achieved in order to be able to increase performance in the long term. Furthermore, an attempt is made to explain why some training contents seem to be rather unsuitable when it comes to increasing strength as a basis for higher performance in the start, turn and clean swimming. Practical training consequences are then derived from this. Regardless of the athlete’s performance development, preventive aspects should also be considered in the discussion. The article provides a critical overview of the abovementioned key issues. The most important points when designing a strength training program for swimming are a sufficiently high-load intensity to increase maximum strength, which in turn is the basis for power, year-round strength training, parallel to swim training and working on the transfer of acquired strength skills in swim training, and not through supposedly specific strength training exercises on land or in the water.

## 1. Introduction

The importance of strength or strength training for swimming performance has been discussed since the early 20th century. It is particularly associated with Robert Kiphuth, who was likely one of the first swimming coaches in the 1920s–1930s to implement training outside the pool (dry-land training) in the attempt to strengthen the muscles relevant to swimming performance [1,2]. The importance of strength training for swimming has been and still is the subject of debate, and its ability to impact swimming performance is often underestimated by some authors [3]. However, in addition to the performance gains associated with strength training, it is important to consider the injury preventive aspects of integrating strength training into the swimmers’ preparations [4]. Obviously, keeping the swimmer healthy is the primary aim but is also a fundamental prerequisite for the realization of the training plan and, based on this, a high level of performance.

However, based on the available literature on strength training and swimming, there are contradictory approaches and several arguments for different strength training methods, and their meanings are only partially empirically justified. For example, Morouço and colleagues [5] write that it is unclear from a scientific perspective whether strength training helps to increase swimming performance and how such training would have to be designed to optimize performance. Questions about the correct periodization of strength training over the course of the season or the annual training plan are also often completely disregarded. Consequently, there are no concrete recommendations for how strength training should be integrated into the individual season contained within the annual training plan for swimmers. A particular problem for the planning of training in swimming is the development of performance-relevant strength (e.g., maximum strength, rate of force development) in combination with predominantly endurance-oriented water-based training activities because extensive endurance training can negatively influence optimal strength development [6]. Although simultaneous strength and endurance training provide competing stimuli that trigger differing adaptive mechanisms [7,8,9,10,11], studies on other endurance sports, such as triathlon [12] and cross-country skiing [13], have reported that strength training can be used successfully despite high volumes in endurance training. This problem also applies to the large training volumes that are commonly undertaken in competitive or high-performance swimming. However, a pure volume orientation in endurance training is no longer considered contemporary [14,15,16,17], which also applies specifically to swimming [18,19,20,21,22,23,24,25,26,27,28,29]. Several research groups emphasize and/or provide evidence that supports the importance of anaerobic energy supply for competitive swim distances up to 200 m [18,30,31,32,33,34]. The exclusive focus on training volume, which is reportedly up to 110 km per week [35], still prevails in some national swim programs. However, such an approach must be questioned, as it does not consider the metabolic requirements of the different competition distances. Costill and co-workers [24] (p. 376) write on this problem:
“Since the majority of the competitive swimming events last less than 3 min, it is difficult to understand how training at speeds that are markedly slower than competitive pace for 3–4 h·d^−1^ will prepare the swimmer for the supramaximal efforts of competition”.

The success of an accompanying strength training program is likely to depend primarily on the optimal combination of the different strength training and endurance training strategies integrated into the swimmers training program, whereby it is not possible to avoid interactions due to the large number of performance factors to be triggered. In the following article, we firstly provide an evidenced-based overview on the importance of strength training for swimming. This is followed by the basic requirements for the design of strength training so that it can produce the desired adaptations.

## 2. Aims of Strength Training in Swimming

The primary objectives of integrating strength training into the preparation practices of swimmers is to prevent the degenerative changes in the active and passive musculoskeletal system and the improvements in various strength parameters (e.g., maximum strength, rate of force development) that influence competition performance. In addition to the influence of strength training on the generation of impulses in the swimming movement, improvements in performance at the start and during the turns are of particular importance for competition success.

### 2.1. Preventive Aspects of Strength Training for Swimmers

The need for early and regular strength training arises from the observation of overuse injuries typical of swimming. The regions of the body affected are primarily the spine [36,37,38,39,40,41,42,43,44,45,46], shoulders [35,36,37,38,39,40,41,43,45,47,48,49,50,51,52,53] and knees [36,37,38,40,41,43,54,55,56]. The causes of these issues are, in particular, orthopedically unfavorable movement sequences (e.g., the leg kick in breaststroke with the consequences of overtraining the medial collateral ligament and/or chondromalacia of the patella, medial compartment synovitis, inflammation and fibrosis of the synovial plica; shoulder: subacromial or intra-articular impingement, reduced glenohumeral stability; spine: degenerative disk changes), incorrect techniques (e.g., stretched arm guidance in the recovery phase in crawl and dolphin swimming, increased lordosis during dolphin swimming) and incorrect use of training aids [4]. Various authors link the use of paddles to the occurrence of shoulder injuries [41,49,52]. The following circumstances are associated with these occurrences [35,44,49,50,51,57,58]: a high volume of training in the water; early entry into the sport; strength training on land with incorrect technique; a rapid increase in swim training volume; a dependence on the styles swum (primarily breaststroke and dolphin); exercises in the water that lead to increased lordosis of the spine while using assistive devices.

While several researchers have reported that the bone structure of competitive swimmers does not differ from that of untrained individuals [59,60,61,62,63,64,65,66,67], others report significantly impaired bone structure (e.g., bone mineral content, bone mineral density) [68,69,70,71]. Various studies have reported that adolescent and adult swimmers have a lower bone mineral density than athletes from weight-bearing and strength-based sports [59,60,64,72,73,74,75,76,77]. These impairments primarily affect the lumbar region of the spine and the lower extremity. Degenerative changes in the spine have been documented by several research groups [57,78,79,80]. For example, Kaneoka and colleagues [81] have reported that about two-thirds of all swimmers studied demonstrate degenerative changes to the spine, which is often associated with self-reported back pain. One factor, which may be associated with reductions in bone density issues, is likely related to the high proportion of weight-relieving training in the water [78]. In this context, the positive influence of strength training on bone structure should be noted [62,64,65,75,82,83,84,85,86,87,88,89,90,91,92,93]. Accordingly, resilience can be increased by influencing the bone structure at an early stage through regular strength training. Apart from classical strength training, any form of “high impact” loading, such as plyometrics, is recommended [61,62,63,67,75,91,92,94,95,96,97,98]. Both the load intensity and the load volume are of great importance for the development of the bone structure [59,60,61,99,100,101,102,103,104].

In addition to the positive effect on bone mineralization, strength training can also improve the stability of knee, hip and shoulder joints. Better joint control could lead to a reduction in joint irritation. However, it is important to note that strength training must be carefully planned and integrated into the training process. If strength training is simply added to the existing training volume, this would lead to an increase in the total training load and consequently increase the risk of overtraining. As such, strength training should not be started if the swimmer displays any sign of overreaching or overtraining. If strength training is incorporated into the swimmer’s training plan, it is important that the remaining swim training volume must be adapted (significantly reduced) to account for the new training content. It is important to note that, from a preventive point of view, strength training should be started early in the athlete’s long-term development plan as starting strength training before puberty can ensure the athlete establish good bone structures [62,82,98,105,106,107]. In the case of shoulder problems, training must be critically analyzed, especially where work is carried out against increased resistance from joint angles that are difficult to stabilize, which is often the case with so-called “specific” strength training exercises on cable traction devices (e.g., biokinetic swim bench).

### 2.2. Strength Training to Increase the Strength Abilities of the Muscles Used to Propel the Swimmer

In addition to injury prevention effects of strength training, it is also important to consider the performance benefits of this type of training. Special attention should be paid to both swim starts and turns, as well as the swimming movement itself as this can also benefit from strength training. By increasing the total impulse, resulting from increasing the partial impulses of the arms and legs, the propulsion speed can be increased.

#### 2.2.1. Increase in the Impulse of Swimming Movements

##### What Should Be Called Strength Training?

From a biomechanical point of view, an increase in swimming speed can be achieved in two ways. Firstly, this can be achieved by optimizing the cycle frequency and/or lengthening the swim stroke. There is, however, a distance-dependent, optimal relationship between cycle path and frequency, as an increase in frequency can lead to a reduction in cycle path and vice versa. Lengthening of the cycle path can be achieved in two ways: firstly, by reducing the braking force (negative acceleration: e.g., inhibiting water resistance) and secondly, by increasing the propulsive forces. Strength training can positively influence both the cycle frequency and the cycle path (by increasing the propulsive force) [108,109]. The extent of the effect of strength training depends on the level of performance and the competition distance. To increase the overall propulsive force is the result of increasing the force of a single movement, which can be achieved by developing maximum strength. In addition, in the case of repetitive cyclic loading over time, the reduction in impulses must be kept as low as possible. This is mainly carried out by training the competition-specific metabolic situation. The term “strength endurance” is often used in this context [110]. Strength endurance refers to the ability of the neuromuscular system to realize the highest possible sum of impulses during a given time period against higher resistances.

Impulse:(1)$$P_{Total}=\sum_{i=1}^{n}\int_{t_{i1}}^{t_{i2}}F_{i}(t)\;dt$$

Impulse consists of the magnitude of the single force impact and the ability to keep the reduction of these force impacts as low as possible (fatigue resistance). It should be noted that the exact border at which strength endurance is distinguished from endurance performance is not clearly defined in the literature. This often leads to supposedly contradictory statements regarding the role of strength endurance in swimming.

The term strength training should only be used when central nervous adaptations, associated with a high degree of activation (nearly complete activation of the motoneuron pool in a short time frame) and/or morphological adaptations are the long term training goal. Such adaptations are typically linked to the use of high intensity training loads (relative to maximum strength (1 RM)). For the untrained, relatively low load intensities, which should not fall below 50 to 60% of the 1 RM, these are initially sufficient. However, no positive adaptations (e.g., for improving bone structure) are to be expected at low load intensities, even for the untrained [91]. Due to the high energy demand and the fact that at intensities above 50–60% of maximum strength, blood flow to the muscle is severely impaired [111,112], the energetic demands are primarily supplied by anaerobic metabolism [85,113]. The further the training load deviates from an intensive activation of the musculature by the central nervous system and a dominant anaerobic energy supply to a more frequent and lower training load, the lower the contribution of strength training to performance. After a maximum of two to three minutes, it can be assumed that aerobic energy supply dominates [114,115,116,117]. A distinction between strength and endurance training is physiologically difficult to justify and is therefore always arbitrary. However, it makes sense to assign training loads with force inputs of less than 50–60% of the maximum force and thus dominantly aerobic metabolic state to endurance training, as they do not lead to neuronal and morphological adaptations that are characteristic of strength training in the long term [118,119]. However, it is important to note that at the beginning of strength training, even lower intensities can be effective for a few months [120,121]. Training with lower intensities tends to lead to long-term metabolic adaptations, which are probably better developed with swim-specific training in the water. When considering training intensity, it can be assumed that in the long term, use of intensities below 80% of the 1 RM does not further enhance the active (muscle) and passive (e.g., bone) musculoskeletal system, which are often considered to be primary goals of strength training [122]. This also explains why training with high numbers of repetitions and low intensities does not further enhance strength gains after a few weeks and months of training and therefore should be considered as ineffective for enhancing high level performances. Based on this line of reasoning, strength endurance training is not advisable for swimmers, even when incorporated as a method of training variation. For athletes with low strength levels, a positive effect of training with low loads on the stroke frequency (i.e., an increase in the number of power strokes per time unit) can be expected over a short time period. This is particularly relevant for effective footwork, at leg stroke frequencies of over 120 cycles/min [123,124,125,126]. For swimmers with higher strength levels, a maximum strength increase may result in the ability to sustain a stroke frequency for a longer duration and facilitate an increase in distance gained per single stroke. All three effects (higher stroke frequency, sustaining a stroke frequency and increased stroke length) would contribute to an increase in the sum of impulses. However, as mentioned earlier, the performance gain from strength training is influenced by the number of cycles required to complete the distance to be swum.

There is a paucity of literature that has examined strength training with sufficient amounts of higher intensity training that is needed to cause long-term improvements in maximum strength. Most interventions either involve working against increased resistance in the water [127,128,129,130], the attempt to simulate the swimming movement with increased pulling resistance on the biokinetic swim bench on land [127,131,132], or performed training with low resistances and high numbers of repetitions [131,133,134,135,136,137,138,139]. However, due to the duration and the low intensity of the training load, such procedures are typically classified as endurance training. In this context, it is often said that this is specific or semi-specific strength training. Other studies do not allow a clear assessment of the training method [108,129,140]. Exercise selection is another problem in some investigations. It is often not possible to assess whether the selected training exercises even allow for the application of high resistances. For example, training on an unstable surface or exercises that are performed at joint angles that cannot be sufficiently secured, limit the resistance that can be used while training [141]. In this case, stated intensities and low numbers of repetitions lead to the believing of a high-load intensity. In other words, the postural challenge leads to a false sense of high intensity. To keep the risk of injury low, even the determined maximum force may represent a resistance that is too low to provoke a training stimulus in such conditions, which makes the training effectiveness of these type of interventions questionable. It is noticeable that in swimming, all training content that allows a greater impulse to be generated than is possible in the water is often referred to as strength training. However, it is questionable whether this is a sufficient criterion to give a training intervention the name strength training. Analogously, for a marathon runner, any form of training in which a greater impulse is generated than in the marathon itself would also be referred to as strength training if this definition were used. Strength training, as the name suggests, should target the development of strength or maximum strength, and thus includes specific physiological adaptations. If one follows this approach, most of the training interventions used in studies that are intended to increase maximum strength on the one hand and swimming performance on the other are intensive endurance training and not strength training. Therefore, these studies cannot be used to answer the question of the effectiveness of strength training for increasing swimming performance.

##### Cross-Sectional and Longitudinal Studies on Strength Training in Swimming

For swimming performance without consideration for starts and turns, the musculature of the upper extremity is of great importance [142,143,144,145,146,147,148,149,150] and the musculature of the lower extremity is of a secondary importance [18,142,146,151,152,153]. However, the available evidence is not consistent and does not consider the different styles of swimming. The opposite is evident for the literature on the start and the turn. Correlations between swimming speed (10 m to 50 m distance) or times swum and maximum strength values range between r = 0.49 and r = 0.88 [109,128,145,154,155,156,157,158,159] and r = −0.41 and r = −0.80 [160,161,162,163,164,165,166]. Additionally, these correlations have been confirmed in children (10–14 years) [167]. Moderate correlation coefficients with swimming performance (freestyle, 15–25 m) can also be shown for rate of force development (RFD) [159,168]. The RFD is influenced by maximal strength, but also other physiological factors (e.g., muscle mass, fiber type composition, motor unit discharge rate, muscle-tendon stiffness) [168,169,170,171,172,173,174]. For maximum physical power, which also depends on maximum force generating capacity [166,173,175,176,177,178,179,180,181,182,183,184,185,186,187,188,189,190,191,192,193], high correlation coefficients (r = 0.63 and r = 0.9) are reported for distances between 22 m and 400 m [33,148,149,158,194,195,196]. The extent of the relationship between the ability to express high power outputs and swim performance appears reliant on the distance swum [149,196,197]. While strength is generally associated with distances < 400 m, several authors have suggested that there is a positive relationship between strength and performance during distances >400 m [141]. For example, due to the positive relationship between maximum strength and swim performance, the increasing number of turns associated with longer distances could explain a positive influence of strength on performances at these distances [198,199,200]. The usual problems associated with the analysis of this data must be considered with these correlations. While the justification of the causal relationship of the parameters is straightforward, the measurement methodology, the performance level of the athletes and the heterogeneity of the sample remain points of criticism. Apart from the inevitable question of whether data collected on swimmers with an average to good performance level can be transferred to elite swimmers, the other two points pose a problem for the calculation of correlations in some studies. For example, a performance heterogeneous sample promotes high correlation coefficients, while a homogeneous group (e.g., a group of top athletes) conceals possibly existing correlations. With regard to the collection of force data, it should be emphasized that isometric measurements are often used, which may not realistically represent the maximum strength. The basic problems here are the joint angle at which the test is carried out and the type of testing (e.g., single or multi-joint, positioning of the athlete). Difficulties in transferring maximum isometric values to dynamic conditions are well known in the literature. Jones and Rutherford [201] were able to show that after dynamic strength training, an isometric measurement of maximum strength does not correctly reflect the value determined under dynamic conditions. The isometric measurement of maximum force led to a significant underestimation of the real maximum force change [202]. Nuzzo and colleagues [203] confirmed this with EMG (electromyography) derivations on the erector spinae muscle. The authors provide evidence for a significantly lower activation of the target musculature in maximal isometric work compared to the activation in the squat or deadlift. This leads to an underestimation of realistic performance capacity during exercises in which the muscles cannot be activated to the maximum. Of course, this is not only true for isometric testing, but for any unfamiliar testing condition. McGuigan and Winchester [204] and Haff and colleagues [205] state a correlation between isometric and dynamic maximum strength values of r = 0.61–0.80. It can be deduced from this that it is questionable to carry out a test on specific angles or to select a working method of the musculature that is not known to the athlete or has not been trained. Numerous longitudinal studies have shown that this can lead to serious misinterpretations of the strength training effects determined [201,206,207,208,209,210,211,212,213,214,215,216,217]. First of all, what was trained should always be tested. Beyond that, careful consideration must be given to what additional tests may be helpful. This is therefore not simply a research methodological problem, but should also take performance diagnostic measures into account.

As already mentioned, the analysis of published longitudinal studies also reveals various research methodological problems. The two most serious are the short duration of these studies [218,219] and the attempt to make strength training specific. The higher the performance level in the sample, the less likely it is to achieve an improvement in swimming performance when training is undertaken for short periods of time, such as a few weeks [219]. However, researchers are usually limited in their access to athletes, as the amount of time available is dictated by competition schedules and the resulting training plans. The attempt made in some studies to make strength training specific should also be examined much more critically, since only purely biomechanical or phenomenological characteristics are usually used as criteria for specificity. The demand for a kinematic and kinetic match, or at least the approximation of a strength training exercise to a sport-specific technique, often lacks consideration for physiological processes. It is then usually a purely phenomenological observation with the intention of imitating a movement sequence in the weight room, without considering that physiological processes are responsible for the creation of the movement. If one considers the discussion about the most effective realization of propulsive forces and thus about the best swimming technique [220,221,222,223], such suggestions of specificity must be even more surprising. It must be clear that resistance training can only approximate a target movement, since movements performed with higher external resistance, through the use of additional loads, always result in a change in the kinematic and kinetic characteristics of a movement. The idea of choosing or creating exercises that are similar to the target movement(s) of swimming is problematic in terms of the CNS (central nervous system) activation of the muscles. The desire to align strength training exercise and sport-specific movement as much as possible is understandable, as an increase in performance in the weight room does not directly translate into an increase in performance in swimming, especially at a high level of performance. As previously mentioned, this is a problem associated with many longitudinal studies. While at a low performance level the transfer of increased strength abilities into sport-motor tasks is highly likely and a positive influence on athletic performance can be easily demonstrated, this effect becomes less and less pronounced as the athlete’s performance increases. The higher the performance level of the subjects, the longer the training intervention should last in order to increase the probability of a positive training effect.

In many studies, it has been reported that an increase in maximum strength after a training period could often only be determined for conditions that corresponded to those of the training. Several researchers have reported that there are specific adaptations that occur at the joint angles selected during training [214,224,225,226,227], movement speeds [210,228,229,230,231,232,233,234,235,236] and types of contraction [201,208,210,212,234,237,238,239,240,241,242,243,244,245,246]. These results are supported by numerous EMG findings [235,238,242,247,248,249,250,251,252,253,254,255,256,257]. The results show that even small changes such as changing a joint angle lead to deviations in the EMG signal. More generally, increases in strength after training periods are partly linked to motor tasks, such that an increase in strength achieved in one exercise is not necessarily seen in another exercise that uses the same muscles [244,258,259]. Initially, this seems to justify the demand for consistency of biomechanical characteristics. In this context, the researchers often refer to the problem of transferring developed strength abilities into different target movements or performances [258]. Classic studies that demonstrate the transfer of effect problem are generally based on the works of Thorstensson and Rutherford [202,260,261]. Based on their research, merely changing the way a muscle or muscle group works or switching to a different strength training exercise for the same muscle group can mean that the strength gains that are achieved in the training exercise are not easily transferred to another exercise or movement pattern. Many other studies have confirmed this poor transferability of increased performance in the training exercise to a test criterion that did not correspond to it. These include the fact that in several studies, little or no transfer to strength measurements under isometric conditions was found in dynamic strength training [201,202,206,207,208,209,210,211,212,213,214,215,216,217], a problem often not sufficiently taken into account in studies and in performance diagnostic measures. The fact that the CNS is primarily responsible for this lack of transfer to different test criteria could be shown in studies in which an increase in neuronal input, analyzed via EMG, could not be found in all test conditions affecting the trained muscles after a training period [262,263]. The results show that even small changes in a movement cause specific reactions of the CNS.

Strzala and Tyka [33] have reported that a difference in swimming speed of 0.1 m/s leads to a change in biomechanical parameters, which must inevitably lead to a change in muscle activation. Fundamentally, the demand to select or modify strength training exercises in such a way that the activation behavior of the CNS during the execution of an exercise approximates the innervation pattern of the target movement as closely as possible is not realistic, because this approximation to the kinematic (distance–time relationships), kinetic (force–time relationships) and rhythmic movement characteristics of the target movement cannot be realized for most movements typical of the sport, and “approximated” always means “different”! The lack of correspondence between kinetic and kinematic characteristics of the swimming movement and an inevitably resulting altered activation via the CNS are confirmed by numerous authors [150,264,265,266,267,268]. This applies not only to strength training exercises, but also to swimming with pulling resistance [265,266,269,270] and work with paddles [32,270,271,272,273,274,275,276,277,278]. For work on a swim bench, neither the innervation behavior [21,150,264,267,277] nor kinetic or kinematic aspects of the movement [150,267,268] on this training and diagnostic device correspond to those of swimming. Clarys [143] (p. 20) writes:
“It can be stated that there is little electromyographic similarity between swimming movements on dry land and the front crawl movement under normal conditions …”. Bradshaw and Hoyle [151] (p. 15) add: “A limitation of the bench is that most swimmers, in order to produce as much power as possible, use a different pulling technique than it is used in the water. The technique most often used for producing a maximum power measurement on the bench is likely to be less efficient in the water.”

Training with the swim bench should rather be described as intensive endurance training on land. Thus, correlations with different swimming performances are not surprising [149,279]. Costill and co-workers [23] showed that in swimmers, different ways of recording mechanical performance were unrelated, which calls into question their validity. An analysis of longitudinal studies casts doubts on the effectiveness of the swim bench as an effective training tool [23,127,131,132]. However, the number of studies relevant to this topic is too small to be able to make a precise statement.

As a logical consequence, it is often pointed out in swimming literature that “specific” or “semi-specific” exercises can lead to an undesirable change in swimming technique that can interfere with both the acquisition and maintenance of technique [150,265,266,270,273,278]. Uebel [150] (p. 40) writes:
“The swim bench, for example, could create such a negative interference, since it is similar but cannot copy the real movement, which is affected by slippage, drag forces, and the use of the lower extremities”.

It is also astonishing that according to Aspens and Karlsen [18], for example, the effectiveness of paddles for building up the swimmer’s performance has not been proven, so that their use and that of fins must be viewed critically, as movement patterns can undergo significant changes compared to swimming under competition conditions [274]. The degree of similarity between strength training and swimming is limited by the fact that the respective exercise must primarily retain its effectiveness in building a high maximum strength level. Only then can one consider which exercises best ensure the transfer of the acquired strength ability into the target movement. Plisk [280] (p. 342) writes about the attempt to simulate target movements of a particular sport in the weight room:
“Likewise, we must avoid falling into the simulation trap (i.e., being fooled by outward appearances or kinematics). An exercise may look like a target task without being specific to it.”

This is complemented by Vorontsov [281] (p. 324) with the statement:
“The main conclusion following from research data is that since land exercises cannot accurately reproduce specific neuro-muscular patterns of swimming motions the best way to develop specific strength in swimmers would be to work on it during swimming training.”

There are only a few intervention studies in which the training methodological procedure can most readily be classified as strength training [109,165,282,283]. In some studies, it is not clear how the strength training was implemented, as the training parameters or the exercise selection are not clearly described [108,129]. Although the training parameters and the selection of training exercises are designed in such a way that a long-term increase in strength potential would be possible, the existing literature is inconsistent and does not yield definitive evidence that this is indeed the case. A positive influence of a training intervention with the aim of increasing strength has been reported in some studies [108,109,129,165,283,284]. However, in two studies there was no significant difference found in performance development between a combined swim and strength training compared to a group that only performed swimming training [108,129]. In this context, Amaro and colleagues [133] provide evidence that it may take some time for the newly acquired strength to become effective in the swimming movement and thus testing directly after a strength training block does not provide valid information on whether the strength training leads to an increase in performance. Especially when examining athletes/subjects who are inexperienced in strength training, the fatigue effects associated with training can initially override the positive training effects associated with strength training. In addition, Potdevin and colleagues [284] and Garrido and colleagues [285] reported that strength or plyometric training can be easily integrated into the training process in children. Based on the results for both studies, an increase in performance in different jumping, strength and swim tests may occur. However, swimming performance did not change significantly compared to a control group. This contrasts with the results from other studies where no positive influence of strength training on swimming performance could be detected apart from the fact that the researchers could determine a significant increase in strength [282,286]. In summary, it must be stated that there are only a few meaningful studies on the topic of strength training in swimming. Most studies on dryland training fall into the category of endurance training on land with the help of a simulated but non-specific swimming movement. In addition, it is often noticeable in the interventions that can be assigned to strength training that the idea of a “specific” movement must have played a role in the exercise selection. However, the exercise selection should be oriented solely towards a long-term effective increase in strength potential.

#### 2.2.2. Increase in Momentum at Start and Turn

The importance of starts and turns is emphasized and documented by numerous authors, especially in competitive distances up to 200 m [32,287,288,289,290,291,292,293,294,295,296,297,298,299]. Due to the high number of turns in the 800 m and 1500 m races, it also appears that strength levels may have a significant impact on competition performance over longer distances [198,199,200]. Start (>4.65 ± 0.24 m/s) and turns (2.6 ± 0.19 m/s) represent the situations in the race where the highest speeds are achieved compared to free swimming (1.78 ± 0.06 m/s) [300,301]. It is equally true for both competition situations that the greatest possible impulse must be generated in a short moment by the leg and hip extension. For starts, push-off times from the block are given between 0.5 and 0.9 s [289,302,303,304,305,306,307,308], while contact times for turns are usually measured between 200 ms and 600 ms [160,291,309,310,311,312,313,314,315,316]. These very short contact times are in part, as for example in Lyttle and colleagues [310], a result of the concentric push-off phase. The strong forward lean of the upper body during the start causes a small hip joint angle at the beginning of the take-off movement. From this, the upper body is accelerated by an accentuated opening of the hip joint angle. The share of the leg muscles and the hip extensors in the total impulse depends primarily on the swimmer’s starting position. The smaller the knee and hip angles chosen, the longer the acceleration distance and thus the possibility of applying force to the body mass for the purpose of acceleration. Small angles place higher demands on the conditional ability of strength and increase the time on the starting block as well as the contact times during turns. If the necessary strength is available, longer start and turn times can lead to a higher take-off speed at the start or push-off speed at the turn. While Nicol and Krüger [313] and Takahashi and colleagues [316] report a correlation of r = 0.83 between the push-off speed and the momentum generated when pushing off the wall, Cronin and colleagues [317] could not confirm these results. Blanksby and co-workers [160] identified peak force as the best predictor of time to 5 m post-turn using multiple regression results, while Cronin and co-workers [317] found a relationship between countermovement jump height and speed between 2 m and 4 m post-turn. Similar results were obtained by Jones and co-workers [318] who found both significantly better performance in the squat jump (with and without additional load) and significantly better turn times in elite Australian swimmers compared to a lower-performing group. Keiner and co-workers [163] found a correlation between 5 m time and 1 RM in the deep squat of r = −0.54 and the CMJ of r = −0.75.

For the start, correlation coefficients with lower extremity maximal strength values and time to reach the 5 m, 10 m or 15 m mark ranged from r = −0.47–−0.78 [163,166,308]. However, Garcia-Ramos and co-workers [319] found no significant relationship between performance at the start and maximum voluntary isometric knee extension and flexion. De la Fuente and co-workers [320] reported that men can generate higher horizontal forces on the starting block compared to women, resulting in significantly higher horizontal velocities. Comparable results are reported by Slawson and colleagues [307]. Mason and colleagues [306] reported reaction forces on the starting block that in some cases reach the level of twice the body weight. Correlations of r = 0.50 and r = 0.76 were found between measurements of maximum power output in different strength tests and different starting parameters [319,321,322]. In addition, Miyashita and colleagues [322] were able to determine a correlation of r = −0.68 between the maximum power output during an extension in the knee joint and the time to 15 m. In numerous studies, the countermovement jump was compared to the performances at the start. While jump performance correlated with times to 5 m, 10 m and 15 m with r = −0.49–r = −0.85 [163,166,319,323], correlation coefficients between r = 0.57–0.70 [319,324,325,326] have been determined between different strength and performance parameters measured at the starting block.

The few longitudinal studies that have been conducted regarding improvements in performance during starts and turns do not provide consistent results. For example, Lee and colleagues [327] and Breed and Young [324] found no improvement in take-off performance after performing purely plyometric training or a combination of strength training and plyometric training. Jones and co-workers [328] were unable to identify any improvement in turning performance in high performance swimmers after six weeks of either strength or plyometric training. In particular, the results from Breed and Young [324] are surprising, as their subjects were somewhat untrained in relation to their strength abilities (average value in the CMJ was 27.3 cm). The research group led by Hohmann [329], on the other hand, demonstrated positive effects after a four-week strength training intervention, without giving precise details of the training content. Several research groups were able to demonstrate the positive effect of plyometric training on performance during starts and turns [284,326,330,331]. Ruschel and colleagues [332] emphasize that improved physical prerequisites leading to a higher momentum on the starting block are only helpful when the improved push-off from the block transitions into an optimal diving phase. This is supported by results from Cossor and Mason [333], whose data underline the importance of the post-dive phase for the 15 m start time. It is unclear why only jumps were used as training and diagnostic tools in most studies. First, it is surprising that in most studies, jumps are used as a “related” movement or supposedly “specific” test forms. Here, too, one encounters the misunderstandings already described above when assessing what can be considered specific from a CNS perspective and what cannot. In addition to its proximity to the target movement, the chosen jumping form should serve as a means for estimating and developing power. However, as maximum strength is a basic determinant of power, it is questionable whether jumping forms are the correct diagnostic and, at least if they are carried out without strength training to increase maximum strength, training methodological approach, especially as recommendations for training practice are to be derived from the results. The dependence of jumping performance on maximum strength has been sufficiently proven. The correlations between the squat jump and the 1 RM range between r = 0.50 and 0.76 [181,334,335,336,337], for the countermovement jump between r = 0.50 and r = 0.93, provided the maximum force measurement is dynamic [166,182,334,336,337,338,339,340,341]. Here, too, the problem of recording maximum force under isometric conditions becomes apparent. In this case, the literature provides both non-significant [203,204,205,342,343,344,345,346,347] and significant results ranging from r = 0.32 to r = 0.82 [346,348,349,350]. Against this background, the increase in maximum strength and its analysis probably represents a better training methodological and diagnostic approach. However, in the few studies that have taken this approach, only low to medium correlations are found. To explain this, one must also carefully pay attention to the methodology maximum strength is recorded. In most studies, maximum strength was operationalized via single-joint, mostly isometric or isokinetic tests, which only depict a part of the musculature relevant to performance during starts and turns and represent a form of contraction that is usually performed in an unfamiliar way when compared to everyday life and sport. This very likely explains the moderate correlations as shown above.

There are contradictory results regarding the relationship between the contact time when pushing off the pool wall and the subsequent velocity. Some research groups report higher velocities with longer contact times [309,310,314], while this was not the case in other studies [160,316,351]. It can be assumed that there is an optimum for the contact time [351]. This is strongly influenced by the maximum strength of the knee and hip muscles. There is also uncertainty regarding the flexion angle in the knee joint [160,291,310,314,316]. Although a small knee joint angle offers the advantage of a larger acceleration path, it places increased demands on the maximum force, since the rather unfavorable force-length relationship of the extensors at this joint angle means that a high force potential is required to accelerate highly from this position. In this context, Mason and colleagues [312] also demonstrate forces during the push-off from the wall that reach a magnitude of up to twice the body weight force.

## 3. Methodical Approach to Strength Training

### 3.1. Morphological Adaptations

Since muscle mass and its activation by the CNS are the central parameters for the development of maximum strength (Figure 1), it is a logical consequence that hypertrophy training must be undertaken before targeting strength development. This training method requires medium to high intensities (intensity = % 1 RM) in combination with a high-load volume. These load intensities are necessary for the generation of the high tension acting on the muscle fiber, which in turn leads to microtraumata of the tissue. Microtrauma is particularly observed in the z-disc region of the myofibrils [352,353,354,355,356,357]. Mechanical stimulation dictates both the extent of myofibrillar damage in trained muscle [358,359,360,361,362,363,364] as well as the extent of hypertrophy-relevant cellular signaling cascades [365,366,367,368,369,370,371,372,373]. These findings [365,366,367,368] are supported by animal studies (in vivo, ex vivo) that point to the importance of high-tension stimuli in triggering muscle hypertrophy [369,374,375,376,377,378,379,380,381,382,383]. The mechanical stimulus generated via high exercise intensities is thus to be regarded as a decisive criterion for the increase in protein synthesis and the resulting hypertrophy of skeletal muscles [365,366,367,368,369,370,371,372,373,374,375,376,377,378,379,380,381,382,383,384,385,386,387,388,389,390,391,392,393,394]. In strength training, particular importance is attributed to the eccentric phase of the movement, which is characterized by a relatively low energy cost, but a very high mechanical stimulus compared to the concentric part of the movement [240,241,361,395,396,397]. Exercise intensities below 60% of 1 RM are too low to produce adaptations in the active or passive musculoskeletal system [91,398,399]. In sports with high strength requirements, even intensities below 80% of the 1 RM are probably no longer sufficient in the long term to trigger further adaptive processes [122,400,401,402,403]. The requirement for a high-load volume can be explained on the one hand by the fact that the extent of intentional tissue damage increases with the performance of several sets for the muscles in training, and on the other hand, by the fact that adaptations in the passive musculoskeletal system, in particular, depend on the load volume, in combination with a sufficiently high-load intensity [86,90,91,404,405,406,407]. As such, the fear that swimmers will rapidly increase body mass is unfounded. In general, it is difficult to build up a significant quantity of muscle within the framework of the high total training volume associated with swimming [11,408,409]. Additionally, to date the negative effect of a pronounced muscle mass on the position in the water and drag force has not been proven [143,410]. Newton and colleagues [410] (p. 7) write in this regard:
“Swimming coaches believe that changes in body shape will increase drag force and this will be detrimental to swimming performance. This contention has not been supported or refuted by scientific research … In truth, the athletes do not have the time to devote to a resistance training program with sufficient volume to produce large increases in muscle size since they complete so many hours training in the pool. It is very unlikely that more than modest gains in muscle size could be achieved in these athletes regardless of the resistance training program. The large volume of endurance exercise that swimmers complete each week is incompatible with maximal gains in strength and muscle size, and past research [408] suggests these conflicting influences will limit muscle hypertrophy.”

Even if it seems logical that an increase in body volume leads to greater water resistance, it remains questionable whether such a change significantly impacts swimming performance. Moreover, any possible negative effects must be contrasted with the positive effects of increased propulsive momentum due to the newly gained musculature. The fact that men swim faster than women despite their greater body mass must be considered when considering the impact of changes in body mass. A positive relationship between swimming performance and body mass or lean body mass has also been demonstrated in adolescent swimmers [411,412,413,414,415,416]. For adults, several research groups have identified a positive relationship between swimming performance and an athlete’s muscle mass [417,418,419]. It is important to note no known research has been able to show a negative relationship between muscle mass and swimming performance. Cronin and colleagues [317] have also reported a positive correlation between body mass and speed 2–4 m after the turn. Chatard and colleagues [417,418] concluded that a larger muscle mass only becomes a disadvantage when the distance exceeds 400 m. For distances up to 400 m, on the other hand, it is an advantage according to the authors [418]. The general fear that an excess of muscle mass will exert a negative effect on swimming performance is likely unfounded. Nevertheless, the degree of skeletal muscle development follows a distance-dependent optimal trend [31]. This means that the longer the distance to be swum, the more disturbing a high body mass becomes. Two to three training sessions per muscle group a week are required to achieve an increase in muscle mass [402], but the effort required increases with increasing performance level [395,396,402,420,421,422,423,424,425]. Hypertrophy training usually causes severe CNS and metabolic fatigue, so that subsequent swimming training should be avoided to prevent negative effects. In addition to an adequate time interval between the training sessions, the content of the water training should be coordinated with the strength training. This strength training method should be used during phases in which moderate load intensities are used in the water.

### 3.2. Development of Strength and Power

To increase the voluntary activation capacity, training stimuli are required that force the motoneuron pool to be activated as completely as possible, and also quickly, if necessary (Figure 1). This requires load intensities higher than 90% of the maximum load. Since the main aim is to improve neuromuscular coordination, the exercises should be carried out in a state of rest and the effects of fatigue during the training session should be kept to a minimum. For the propelling muscles of the upper body, a special consideration for the rate of force development seems to be rather questionable. The reasons for this are:Joint angular velocities in the shoulder joint, which are 240–300°/s [31,426,427],A rather low angular velocity at the beginning of the pulling movement [428],Lower force at the beginning of the pulling phase compared to later parts of the movement [428],A duration of the propulsive pull phase at high swimming speeds of about 400–600 ms [221,222,429,430,431].

According to Schmidtbleicher [432], the explosive force represents the decisive performance-determining factor in movements that require less than 250 ms for their completion.

If more time is available for the development of force, maximum strength increasingly becomes the more important factor. However, other requirements for strength development apply to the lower extremity. Since rapid force development is an important component in starts and turns, strength training should aim to activate as many motoneurons as possible in the shortest time as possible. While with weaker individuals, an increase in maximum strength automatically results in an increase in the rate of force development, stronger individuals simply increasing maximal strength does not result in increases RFD. To prevent misunderstandings, it should be mentioned that a significant increase in muscle activation is already achieved during training that targets hypertrophy, depending on the athlete’s starting level. In swimming, the start and the turns are where a short and intensive activation of the leg and hip muscles is decisive for performance. If the training goal is purely designed to prevent injury, hypertrophy training is the only training method that should be used. When doing this, maximum strength will automatically increase to a certain degree. Due to the high loads and the resulting demands on the trunk muscles and movement technique, training with maximum intensities should only be started after several years of strength training so that the passive musculoskeletal system, in particular, is prepared for exposure to high loads. The use of maximum loads is recommended in phases with lower training volumes. Once again, it should be noted that a high swimming volume has a negative influence on the development of maximum strength and power [6].

Surprisingly, a large number of recommendations for strength training in swimming include training programs that are more similar to the design of a strength endurance training program with lower loads and high numbers of repetitions or a prevention program (e.g., to prevent back pain), the effectiveness of which is questionable depending on the design [433], or an intensive interval training in endurance training [434,435]. This is probably due to the fact that in swimming, any training using impeded (resisted) conditions (e.g., paddles, land training on pull ropes/cables or swimming benches) is often referred to as strength or strength endurance training. This must be surprising, since the strength endurance method—especially in the long term—does not provoke any physiological adaptations that the swimmer would have to develop in the weight room. If one disregards the fact that an untrained person can benefit from almost any unfamiliar training stimulus [219], this changes enhanced adaptability dissipates after only a few weeks of training. In the long term, only adaptations to the metabolic system can be expected when using these types of strength training. If a highly trained athlete (in terms of maximum strength) undertakes strength endurance training, there will be a deterioration in performance after several weeks. Thus, this training intervention should not play a role as a form of variation within the periodized performance build-up. Newton and colleagues [410] point out that the adaptations provoked by strength endurance training are already sufficiently covered by swimming training.

If the athlete is at a relatively low level of performance, it can be expected that performance progress can be achieved with different intensities of training or training methods, since there is always training of coordinative skills and thus an increase in strength [311,409,436,437,438,439]. However, as the level of strength increases, this arbitrariness in method selection disappears.

Surprisingly, when it comes to developing speed strength/power, training with light to medium loads (30–60% 1 RM) is often considered particularly effective. This ignores two circumstances. Most of the studies on which these statements are based were conducted with the untrained or with athletes who have low to moderate maximum strength levels. It is therefore not surprising that this type of training has led to short-term increases in performance in various motor tasks [219]. In addition, power at medium loads is highly correlated with or dependent on maximum force [118,166,173,175,176,177,178,180,181,182,183,184,186,187,188,189,190,191,192,203,440,441]. It is therefore clear that, in the long term, the increase in maximum strength is necessary for increasing power (Figure 2). Against this background, training interventions in swimming that only include jumps, for example, must be viewed critically, as it cannot be expected that such training is effective in the long term regarding the development of power. In addition to maximum strength, the development of the rate of force development is the second decisive factor [441]. In addition, the highest physical power output achieved in a strength training exercise (test exercise) is often erroneously equated with the physical power output in the target movement or movements of the sport. However, high mechanical power output is to be produced in the target motor task of the sport and not in a test condition. Bryant [442] (p. 8) explains:
“Moving quickly with a weight does not mean one will move quickly without a weight. […] It appears that explosive exercises like these use too much resistance to improve speed and too little resistance to increase strength and therefore essentially have no beneficial effect on muscular power or athletic performance.”

O’Shea and O’Shea [31] also point out that the highest possible physical performance must be realized in the swimming movement and the movement speed achieved here, and not in a strength training exercise with medium-load intensities. Lyttle and Ostrowski [152] also write that the selected load intensity in strength training must be high enough to increase maximum strength. It is therefore not useful to use load intensities in strength training that are oriented towards maximum power output. Training with medium loads can at best play a supplementary role in training planning.

In terms of exercise selection, the classic strength training exercises (squat, bench press, lateral pulldown, etc.) should be preferred [427]. They have proven effective in practice for increasing maximum strength and building muscle mass (Figure 3). Exercise selection that suggests a supposedly beneficial “specificity” or “semi-specificity” should be avoided, as this is a fallacy.

Again, Bryant [442] (p. 7) writes very aptly:
“A movement performed using some type of resistance can mimic a particular sport movement, but it can never precisely match the speed and coordination of that movement and therefore cannot be considered specific. A movement is either specific or it is not; it cannot be ‘almost’ specific.”

And is appropriately complemented by Clarys and co-workers [443] (p. 198): “Therefore, it should be emphasized that the ‘dryland’ training against mechanical resistances are nonspecific.”

The transfer of the newly acquired strength abilities into the target movement is still relatively easy to achieve at a low performance level or occurs on its own, as the gain in strength is usually so large enough to, at least partially, arrive in the target movement [444]. However, with increasing performance, it becomes an even greater challenge. This also explains why, at a high-performance level, an increase in maximum strength does not automatically lead to improved swimming performance [31]. Initial training successes achieved with supposedly specific exercises should not obscure the fact that these lose their effectiveness very quickly in long-term performance building.

Zatsiorsky and Kraemer [444] (p. 160) note:
“Perform the main sport with added resistance. This often is the quickest way to make gains in athletic performance. It is also insufficient. The performance results initially advance but soon stop improving due to accommodation. Other training means are then necessary”.

The transfer of newly acquired strength skills must be realized primarily in the water (Figure 4). Starts and turns should be given special attention independent of the rest of the swimming training. In other words, transfer is primarily worked out in the target movements of the sport. In this regard, Sandler [445] (p. 39) appropriately mentions:
“The purpose of the base building phase is not to mimic sport skills, but to allow the entire body to develop and adapt to the stresses of training and competition. The most effective skill transfer occurs when practicing skills as the body becomes stronger, faster, and more efficient. The athlete should strengthen all the muscles used to produce the movement, then practice the movement.”

The feasibility of these suggestions for the design of strength training in performance-oriented swimming sports depends very much on an early start of strength training in the multi-year performance build-up. Haycraft and Robertson [446] recommend that the volume of swimming training should not exceed 5000 m per day if the effects of strength training are not to be impaired too much by endurance training. Already in childhood and youth training, this strength training must be started, as it is highly dependent on how much this part of the conditioning training influences the training of other skills and abilities in later years. Furthermore, seasonal work should be avoided. Only consistent implementation of the guidelines given here can have a positive effect on the swimmer’s performance in the long term.

## 4. Conclusions

The development of maximum strength is the basis for power needed in starts, turns and swimming. The development of strength and power must always take place in parallel with swim training all over the season regardless of the swimming distance. As a general rule, strength training should be carried out at sufficiently high intensities. This is crucial for both morphological and central nervous adaptations in terms of almost complete activation of the musculature in a short time frame. A sufficient load volume in strength training is also important, especially for generating morphological adaptations. The morphological adaptations are of central importance here both from a preventive point of view and as a basis for the development of maximum strength and thus also for the development of power. The load intensity should not be less than 50–60% of maximum strength, even for athletes who are inexperienced in strength training. In the long term, intensities above 75% of maximum strength are advised in order to continue to be able to provoke adaptations. The term “specific” should be used with caution. Swim bench or rope exercises and even swimming with paddles are not specific forms of strength training. Most of these “specific” exercises can at best be seen as strength endurance exercises with little, if any, strength benefit. The transfer of increased strength and power must take place during clean swimming and during the training of starts and turns. In addition, the negative effects of a volume-oriented swim training on the development of strength and power must be taken into account.

## Figures and Tables

**Figure 1 ijerph-19-05369-f001:**
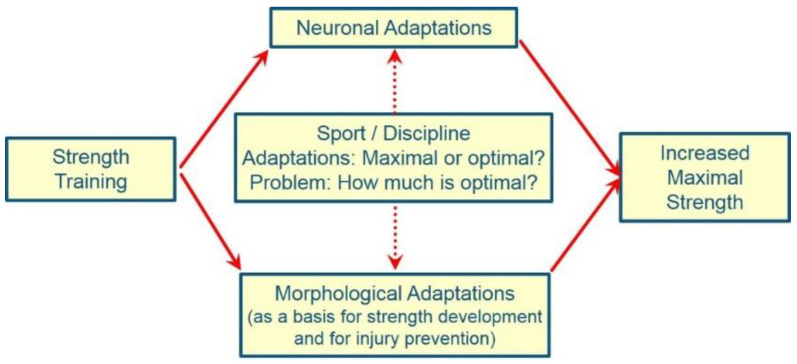
Maximum strength development.

**Figure 2 ijerph-19-05369-f002:**
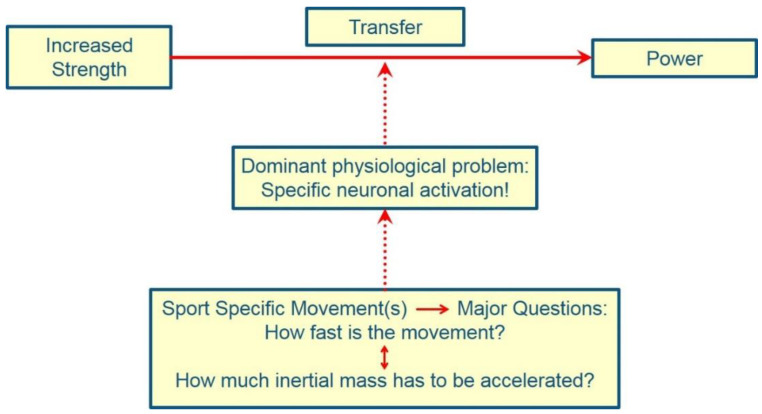
Realization of maximum strength in a fast movement.

**Figure 3 ijerph-19-05369-f003:**
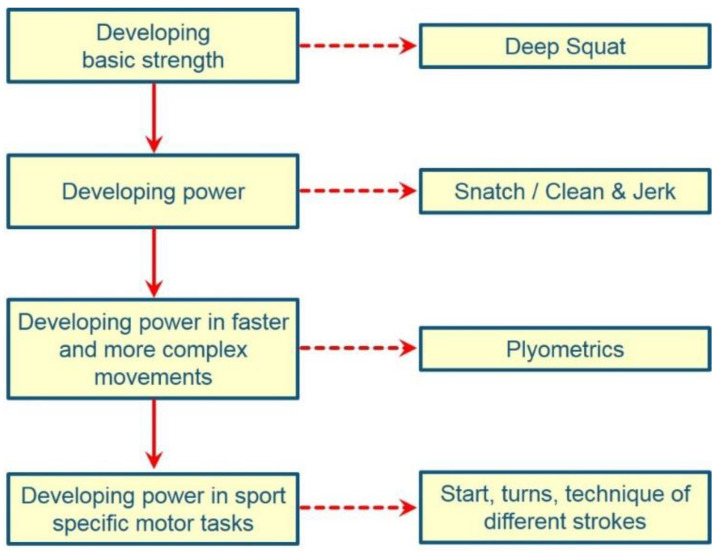
Exemplary notes on exercise selection.

**Figure 4 ijerph-19-05369-f004:**
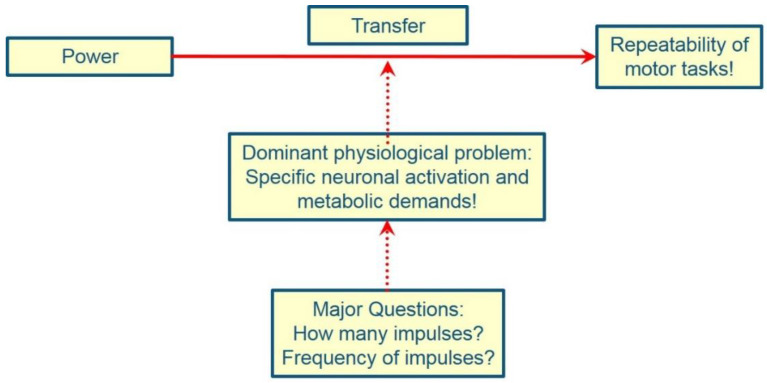
Development of the reproducibility of large impulses in cyclic movements.

## Data Availability

No data reported.

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
