# Peer review of "Strength Training in Swimming"

_ijerph, 2022, doi:10.3390/ijerph19095369_

Round 1
Reviewer 1 Report
First of all, I want to thank you for the opportunity to review this work, as well as congratulate the authors for the quantity and quality of the information provided, closely linked to the topic and scope of the SI.
The purpose of this work is to offer an overview of the literature on different elements that affect swimming performance, as well as to present a proposal to develop them in the long term.
The document is well written, structured and offers a large number of bibliographical references that support the rigor of the work.
Some observations and recommendations are made to facilitate the information to readers in some sections.
1. In relation to the formal structure, it is recommended to check some typographical errors throughout the text, especially regarding the end of the line (for example, line 18, 20, 21, 50, etc; please, check the entire text, although it appears to be a platform bug).
2. In the abstract it is recommended to include some reference to the type of review that has been carried out (narrative review... or similar) so that the reader is clear about the type of work that has been carried out, and a brief explanation so that the researcher be clear about what type of work has been done.
3. A review of those (many) paragraphs that are too long is recommended (perhaps including more full stops and paragraphs), to make it easier for other researchers to read.
4. A final section of the conclusions type is recommended (and would be very useful), since the last section ends "abruptly", and a final contribution (by way of summary) is missing, to synthesize the main findings and proposals .
5. In the same way, and as expressed by the authors in the abstract ("Practical training consequences are then derived from this"), a small final section is recommended in which those derived practical training consequences are exposed.
6. It is suggested that the font of the bibliographical references be adapted to the format proposed by the journal.
Good work!
Author Response
Reviewer 1:
Answer: First of all we would like to thank the reviewers for their work in commenting our manuscript and the time they spent doing this. The notes were helpful, and we edited these clues in our revision
- In relation to the formal structure, it is recommended to check some typographical errors
throughout the text, especially regarding the end of the line (for example, line 18, 20, 21, 50, etc;
please, check the entire text, although it appears to be a platform bug).
Answer: Thank you! That seems to be a problem with the template.
- In the abstract it is recommended to include some reference to the type of review that has
been carried out (narrative review... or similar) so that the reader is clear about the type of work
that has been carried out, and a brief explanation so that the researcher be clear about what
type of work has been done.
Answer: Thank you! Abstract was adapted.
- A review of those (many) paragraphs that are too long is recommended (perhaps including
more full stops and paragraphs), to make it easier for other researchers to read.
Answer: Thank you. We have tried to take this into account in a few passages.
- A final section of the conclusions type is recommended (and would be very useful), since the
last section ends "abruptly", and a final contribution (by way of summary) is missing, to
synthesize the main findings and proposals .
Answer: Thank you! A Conclusion was added.
- In the same way, and as expressed by the authors in the abstract ("Practical training
consequences are then derived from this"), a small final section is recommended in which those
derived practical training consequences are exposed.
Answer: Thank you! We integrated that in the conclusions.
- It is suggested that the font of the bibliographical references be adapted to the format proposed by the journal.
Answer: Thank you! We have to check this because we thought that this is in accordance with the template we received.

Reviewer 2 Report
Dear authors, please find here my considerations and proposals.
Speaking in general, the manuscript is well written and comprehensible for strength training experts, trainers and informed athletes. Congratulations.
The manuscript applies general knowledge of strength training and testing on swimming, which leads sometimes to very specific argumentation and referencing - but its ok.
I agree with your statements to define strength, strength training, specificity of strength training exercises and the problem of test and training similarity having an impact on the interpretation of the available literature making decisions on the effectiveness of strength training in swimming - although the argumentation line is partly very extensive.
Additionally, I appreciate very much that you put your finger on a statistical problem: It is often ignored that correlation coefficients within a sample of homogenous high-level elite athletes cannot reach values as high as among heterogeneous lower-level athletes, thank you.
The number of references is very high; I have to admit that I do not know them all.
I have one major concern:
The text is very extensive, and although the main considerations for practical applications of strength training in the environment of swimming are very well justified, I miss a short, comprehensive conclusion paragraph at the end where the main facts/ recommendations for strength training in swimming are summarized concisely, e.g.
- exercises mimicking swimming on a bench or rope exercises and even paddles are no specific strength training content, but strength endurance exercises with only small strength benefits
- muscle hypertrophy strength training is favourable for injury and overuse prevention
- strength gains need CNS activation, and this in turn needs high intensities of about 80-90% 1RM for elite swimmers
- no block periodization for strength training, but strengthening on-land and swimming in-pool continuously parallel all over the season
- no more than 5.000 km in the pool to avoid interfering of strength training benefits
Maybe, I forgot some key facts. Please think about it.
Those key facts should be touched too in the abstract as an important service for the reader. Thank you very much.
However, I noticed some small points, which I would like to propose for a short reconsideration or rebuttal. Thank you.
Line 78-79 + Line 81: is there an unintended repetition (twice: start & turn)?
Sorry for that.
Line 114-115 - your example "plyometrics". You write "Apart from classical strength training, any form of 'high impact' loading, such as plyometrics, is recommended".
Is it a good example for a strength training mode to promote increases in bone mineral density? Plyometrics with an extremely short 'stretch-shortening-cycle, or should there be mentioned a better example with high loads plus longer time-under-tension? Probably the weight-lifting clean or front squats with higher loads?
A short rebuttal will be ok, thank you.
Line 179-180 - Where do these 50-60% come from? I may be wrong, but I remember a borderline between strength and endurance at about 30% 1RM (dynamic) or 15% 1RM (isometric), e.g. Nicolaus (1995) and Pach (1991).
- NICOLAUS, Jürgen Kraftausdauer als Erscheinungsform des Kraftverhaltens. Köln 1995
- PACH, M. Empirische Untersuchung zur Abgrenzung verschiedener Kraftausdauerfähigkeiten. München 1991
A short rebuttal will be ok, thank you.
Line 183 - typo: affective or effective? Sorry.
Line 216 - typo: ...can be used (?) Sorry.
Line 338 - typo: missing 'full stop' dot. Sorry.
Line 344 - typo: missing 'full stop' dot. Sorry.
Line 429 - typo: missing s "it appears". Sorry.
Line 483 - missing words? ' The strong forward lean of the upper body [during start] causes a small hip joint'. Or am I wrong?
Line 502 - missing words? ' The correlations between the squat jump and the 1RM [in deep-squats] range 502 between. Or am I wrong? 1 RM in another test exercise, or even isokinetic testing?
Line 556 - slash too much? "... swimming \ [401-"
Author Response
Reviewer 2:
Answer: First of all we would like to thank the reviewers for their work in commenting our manuscript and the time they spent doing this. The notes were helpful, and we edited these clues in our revision
The text is very extensive, and although the main considerations for practical applications of strength training in the environment of swimming are very well justified, I miss a short, comprehensive conclusion paragraph at the end where the main facts/ recommendations for strength training in swimming are summarized concisely, e.g. exercises mimicking swimming on a bench or rope exercises and even paddles are no specific strength training content, but strength endurance exercises with only small strength benefits muscle hypertrophy strength training is favourable for injury and overuse prevention strength gains need CNS activation, and this in turn needs high intensities of about 80-90% 1RM for elite swimmers no block periodization for strength training, but strengthening on-land and swimming in-pool continuously parallel all over the season no more than 5.000 km in the pool to avoid interfering of strength training benefits. Maybe, I forgot some key facts. Please think about it. Those key facts should be touched too in the abstract as an important service for the reader. Thank you very much.
Answer: Thank you! We adapted the abstract and added a conclusion!
Line 78-79 + Line 81: is there an unintended repetition (twice: start & turn)?
Sorry for that.
Answer: Thank you, we corrected that.
Line 114-115 - your example "plyometrics". You write "Apart from classical strength training, any form of 'high impact' loading, such as plyometrics, is recommended". Is it a good example for a strength training mode to promote increases in bone mineral density? Plyometrics with an extremely short 'stretch-shortening-cycle, or should there be mentioned a better example with high loads plus longer time-under-tension? Probably the weight-lifting clean or front squats with higher loads?
Answer: Thank you very much! We missed the “volume problem”. We added: “Both the load intensity and the load volume are of great importance for the development of the bone structure (Cassel et al. 1996; Courteix et al. 1998; Dyson et al. 1997; Fehling et al. 1995; Helge & Kanstrup 2002; Lehtone-Veromaa et al. 2000; Nickols-Richardson et al. 1999, 2000; Nurmi-Lawton et al. 2004).”
Line 179-180 - Where do these 50-60% come from? I may be wrong, but I remember a borderline between strength and endurance at about 30% 1RM (dynamic) or 15% 1RM (isometric), e.g. Nicolaus (1995) and Pach (1991).
NICOLAUS, Jürgen Kraftausdauer als Erscheinungsform des Kraftverhaltens. Köln 1995
PACH, M. Empirische Untersuchung zur Abgrenzung verschiedener Kraftausdauerfähigkeiten. München 1991
Answer: Thank you, we added (Schmidtbleicher, D. (2003). Motorische Eigenschaft Kraft: Struktur, Komponenten, Anpassungserscheinungen, Trainingsmethoden und Periodisierung. in: Fritsch, W. (Hrsg.): Rudern - erfahren, erkennen, erforschen, S.15-40.
Line 183 - typo: affective or effective? Sorry.
Answer: Thank you, should be “effective”!
Line 216 - typo: ...can be used (?) Sorry. – Thank you very much!
Line 338 - typo: missing 'full stop' dot. Sorry. – Thank you!
Line 344 - typo: missing 'full stop' dot. Sorry. – Thank you!
Line 429 - typo: missing s "it appears". Sorry. – Thank you!
Line 483 - missing words? ' The strong forward lean of the upper body [during start] causes a
small hip joint'. Or am I wrong?
Answer: Thank you! In line 438 we added “during the start” to make it clearer.
Line 502 - missing words? ' The correlations between the squat jump and the 1RM [in deepsquats] range 502 between. Or am I wrong? 1 RM in another test exercise, or even isokinetic testing?
Answer: Thank you! The 1RM was measured in different ways.
Line 556 - slash too much? "... swimming \ [401-" – Thank you!
Overall: Thank you very much for your close reading of the text.!
